# Molecular basis of antibiotic self-resistance in a bee larvae pathogen

Tam Dang [1], Bernhard Loll [2], Sebastian Müller[1], Ranko Skobalj[1], Julia Ebeling [3], Timur Bulatov[1], Sebastian Gensel[1], Josefine Göbel [3], Markus C. Wahl[2,4], Elke Genersch[3,5], Andi Mainz[1] & Roderich D. Süssmuth [1 ✉]

*Paenibacillus larvae*, the causative agent of the devastating honey-bee disease American Foulbrood, produces the cationic polyketide-peptide hybrid paenilamicin that displays antibacterial and antifungal activity. Its biosynthetic gene cluster contains a gene coding for the *N*-acetyltransferase PamZ. We show that PamZ acts as self-resistance factor in *Paenibacillus larvae* by deactivation of paenilamicin. Using tandem mass spectrometry, nuclear magnetic resonance spectroscopy and synthetic diastereomers, we identified the N-terminal amino group of the agmatinamic acid as the *N*-acetylation site. These findings highlight the pharmacophore region of paenilamicin, which we very recently identified as a ribosome inhibitor. Here, we further determined the crystal structure of PamZ:acetyl-CoA complex at 1.34 Å resolution. An unusual tandem-domain architecture provides a well-defined substrate-binding groove decorated with negatively-charged residues to specifically attract the cationic paenilamicin. Our results will help to understand the mode of action of paenilamicin and its role in pathogenicity of *Paenibacillus larvae* to fight American Foulbrood.

[1] Institut für Chemie, Technische Universität Berlin, Berlin, Germany. [2] Institut für Chemie und Biochemie, Strukturbiochemie, Freie Universität Berlin, Berlin, Germany. [3] Institute for Bee Research, Department of Molecular Microbiology and Bee Diseases, Hohen Neuendorf, Germany. [4] Macromolecular Crystallography, Helmholtz Zentrum Berlin für Materialien und Energie, Berlin, Germany. [5] Institut für Mikrobiologie und Tierseuchen, Fachbereich Veterinärmedizin, Freie Universität Berlin, Berlin, Germany. ✉email: roderich.suessmuth@tu-berlin.de

Pollination of wild and cultivated flowering plants is an indispensable ecosystem service, which is mainly provided by pollinating insects. Among the insect pollinators, managed honey bee colonies play a particularly important role in agriculture, where they are widely used as commercial pollinators and contribute to 35% of the production volume of global food crops[1]. In order to secure human food supply, it is therefore important to ensure the health of honey bees, which is continuously threatened by the overuse of insecticides such as neonicotinoid[2] in agriculture and also by various viral, bacterial, and fungal pathogens as well as metazoan parasites[3].

The Gram-positive, facultative anaerobic, spore-forming bacterium, *Paenibacillus larvae* (*P. larvae*), is the causative agent of the epizootic American Foulbrood (AFB) of honey bees[4]. AFB is the most serious bacterial disease of honey bees and is classified as notifiable disease in most countries because it is highly contagious and lethal to entire colonies. Furthermore, most authorities consider the killing of diseased colonies and burning of the hive material the only workable control measure resulting in considerable economic losses in apiculture. AFB is a fatal intestinal infection of the honey bee brood initiated in first instar larvae by ingestion of spore-contaminated food. The distribution of the spores, the infectious form of *P. larvae*, within a colony and between colonies, also within apiary and between apiaries[5], consequently leads to honey bee colony losses. The use of enterobacterial repetitive intergenic consensus (ERIC) sequence primers has revealed four well-described genotypes ERIC I to ERIC IV[4] for *P. larvae* which differ in virulence on the larval[6] and colony level[7] as well as in pathogenesis strategies employed to kill the host[8]. The existence of another ERIC genotype, ERIC V, has recently been proposed[9]. From contemporary outbreaks of AFB all over the world, only *P. larvae* ERIC I and ERIC II can be isolated[10], suggesting that the hypervirulent genotypes ERIC III to ERIC V did not become established in the honey bee population.

In our quest to find sustainable control measures against this most serious bacterial disease of honey bees, we started to unravel AFB pathogenesis by analyzing the interaction between *P. larvae* and honey bee larvae on a molecular level. We identified several virulence factors of *P. larvae* ERIC I and ERIC II and showed that two AB toxins[11,12], a chitin-degrading enzyme[13,14] and also an S-layer protein[15,16] have a pivotal role in the virulence of this pathogen and that *P. larvae* also produces various secondary metabolites[17]. Bacterial secondary metabolites, with polyketides and (non-)ribosomal peptides as important representatives, provide highly valuable lead structures, among them antibiotics with novel modes of action for drug development to fight various infectious diseases[18,19]. Secondary metabolites can also act as virulence(-like) factors, functioning as signal molecules in gene regulation of defense or growth mechanisms[20–22]. The search for secondary metabolites produced by *P. larvae* led to the structural elucidation of paenilamicin that shows cytotoxic, antibacterial and antifungal activities[23,24]. It is currently assumed that paenilamicin is produced as a defense molecule against microbial competitors, since only *P. larvae* can usually be isolated as a pure culture from the cadavers of AFB-killed larvae, suggesting that other saprophytic competitors are absent in the degradation process of the larval cadavers to the characteristic ropy mass[25]. We recently substantiated this view by showing that paenilamicin was active against the bee-associated saprophyte *P. alvei* in *P. larvae*-infected larvae[23].

Paenilamicin is a linear, cationic aminopolyol peptide antibiotic and is synthesized via an unusual nonribosomal peptide synthetase-polyketide synthase (NRPS-PKS) hybrid assembly line that exhibits several fascinating biosynthetic features. It contains unusual structural motifs such as galantinamic acid (Glm), agmatinamic acid (Aga), *N*-methyldiaminopropionic acid (mDap), galantinic acid (Gla) and a 4,3-spermidine (Spd) at the C-terminus (Fig. 1). *P. larvae* produces a mixture of paenilamicin variants A1, A2, B1, and B2. They only differ in two positions of the paenilamicin backbone: at the N-terminus and in the center between mDap1 and Gla. Either a lysine (series A) or an arginine (series B) is activated by the adenylation domain of NRPS1 (Fig. 1). The amino acid residue between mDap1 and Gla is a lysine (series 1) or an ornithine (series 2) assigned to be incorporated by NRPS4 (*pamD*), respectively (Fig. 1).

The *pam* gene cluster harbors a gene encoding the putative acetyl-CoA-dependent *N*-acetyltransferase PamZ, which belongs to the Gcn5-related *N*-acetyltransferase (GNAT) superfamily[26,27]. One prominent member of this superfamily is the bacterial

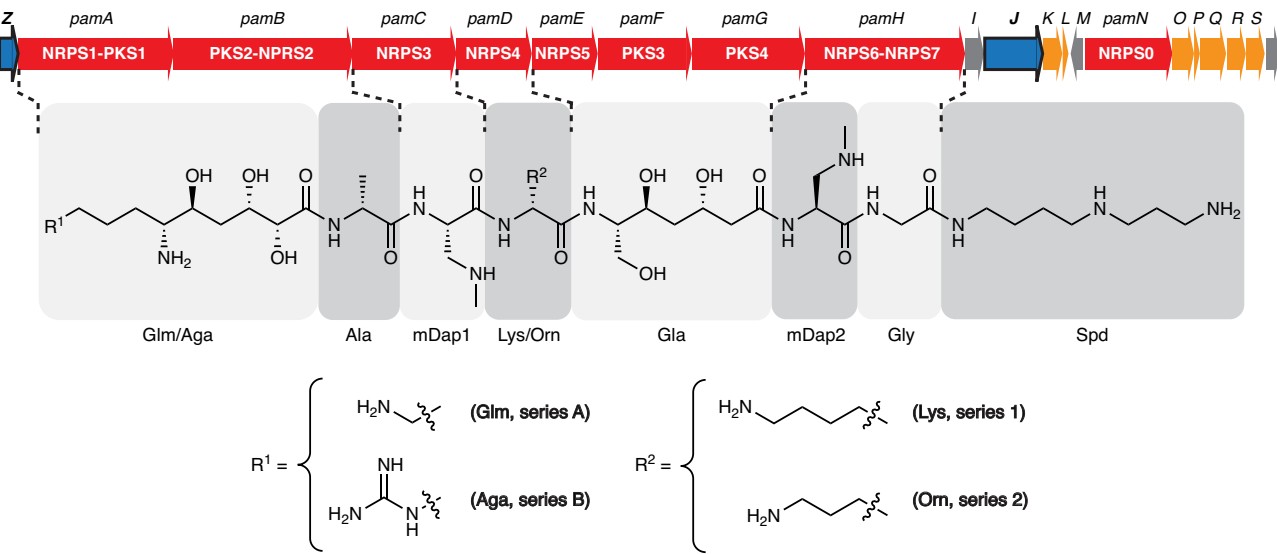

**Fig. 1 Biosynthetic gene cluster and structure of paenilamicin variants.** The paenilamicin (*pam*) gene cluster[23] contains core biosynthetic (red), auxiliary biosynthetic (orange), resistance (*pamZ* and *pamJ*; blue) and other (gray) genes and expresses the nonribosomal peptide synthetase-polyketide synthase (NRPS-PKS) hybrid biosynthetic machinery for the production of paenilamicin A1 (Glm, Lys), A2 (Glm, Orn), B1 (Aga, Lys), and B2 (Aga, Orn). Glm - galantinamic acid, Aga - agmatinamic acid, Lys lysine, Orn - ornithine, Ala - alanine, mDap - *N*-methyldiaminopropionic acid, Gla - galantinic acid, Gly - glycine, Spd - 4,3-spermidine.

aminoglycoside *N*-acetyltransferase (AAC) that plays an important role in antibiotic resistances, particularly in clinical and environmental settings[28]. Aminoglycoside antibiotics have been widely used in the treatment of bacterial infections but they rapidly lose activity against multi-resistant bacteria due to adaptation and the development of resistance. By contrast, self-resistance is an innate, non-adaptation-based mechanism for the protection against self-produced antimicrobial agents. Since self-produced antimicrobial agents could also harm the bacterial host, self-resistance is critical for survival and territorial competition.

Our results demonstrate the deactivation of paenilamicins by the regio- and stereoselective self-resistance protein PamZ including its high-resolution crystal structure that shows how its tandem-domain arrangement may organize substrate binding. Together with a parallel study[29], in which we report on the total synthesis and the biological evaluation of paenilamicin, we have here unambiguously identified the N-terminal building block of paenilamicins as an essential switch for target binding, biological activity and self-resistance.

## Results and discussion

**Regio- and stereoselective *N*-acetylation of paenilamicin by PamZ.** To confirm our hypothesis that PamZ (NCBI WP_023484187) is an acetyl-CoA-dependent *N*-acetyltransferase that targets paenilamicins, we monitored PamZ-mediated antibacterial effects in vitro by agar diffusion assays against *Bacillus megaterium* (*B. megaterium*) as indicator strain as well as by mass spectrometry (MS) and nuclear magnetic resonance (NMR) spectroscopy. To this end, the *pamZ* gene was amplified from the wild-type (WT) *P. larvae* ERIC II strain, inserted into the commercial pET28a(+) vector, and transformed into *E. coli* BL21-Gold(DE3) for heterologous expression. PamZ was then purified (Supplementary Fig. 1) and used for the assays including four native paenilamicin variants as substrates and acetyl-CoA as co-substrate. The paenilamicin variants were purified from *P. larvae* ERIC I and ERIC II, which preferably produce the paenilamicin mixtures A2/B2 and A1/B1, respectively (Fig. 1 and Supplementary Fig. 2). In addition, we also tested synthetic paenilamicin B2 (**PamB2_3**)[29].

The agar diffusion assays clearly showed that paenilamicins incubated with PamZ and acetyl-CoA were not able to inhibit the growth of *B. megaterium*, whereas antibacterial activity was observed in the absence of acetyl-CoA and/or PamZ (Fig. 2). This loss of biological activity correlated with the conversion of paenilamicins to the corresponding *N*-acetylpaenilamicins as observed by HPLC-ESI-MS. ESI mass spectra revealed that the mass-to-charge ratios of natural and synthetic paenilamicins

exhibited a characteristic mass shift of 42 Da indicative of the addition of an acetyl group (Supplementary Figs. 3–7).

Paenilamicin contains several primary and secondary amino groups that are potential candidates for *N*-acetylation. To determine the site of acetylation, we monitored PamZ-mediated effects in fingerprint tandem MS and NMR spectra of paenilamicin before and after treatment with PamZ/acetyl-CoA. Besides the mass shift of 42 Da for the acetylation, characteristic $MS^2$ fragmentation patterns originated from the difference between Glm and Aga residues in series A and B (+28 Da) as well as the difference between Lys and Orn residues in series 1 and 2 (+14 Da). $MS^2$ fragmentation mainly resulted in fragment ions $b_4$, $y_4$ and $y_6$ of each paenilamicin and *N*-acetylpaenilamicin variant acquired by collision-induced dissociation (Supplementary Table 1). Fragment ion $b_4$ varied depending on the paenilamicin series showing mass shifts of 14 Da and 28 Da. Importantly, we observed a mass shift of 42 Da only for fragment ion $b_4$, indicating acetylation in the N-terminal half of paenilamicin. By contrast, the fragment ions $y_4$ and $y_6$ did not exhibit any mass shifts of 42 Da between paenilamicins and *N*-acetylpaenilamicins. Thus, we excluded acetylation in the C-terminal half of paenilamicin (Supplementary Figs. 8–18). In addition, we detected and isolated small amounts of *N*-acetylpaenilamicin A1, B1, and B2 from supernatants of *P. larvae* ERIC I and ERIC II (Supplementary Fig. 19), and compared them with our products formed in vitro. The $MS^2$ fragmentation analysis confirmed that the mono-acetylation in the N-terminal half of paenilamicin also occurred in vivo (Supplementary Figs. 20–22). The $MS^2$ experiments did not reveal whether the N-terminal amino group of Aga/Glm or its side chain (amino/guanidino group) was acetylated.

To ultimately identify the functional group that is modified by PamZ, we acquired $^{1}H$-$^{13}C$ hetero-nuclear single-quantum coherence (HSQC) NMR spectra of paenilamicin B2 before and after incubation with PamZ/acetyl-CoA. Although both spectra were mostly superimposable, severe chemical shift perturbations (CSPs) were observed for a minor fraction of cross-peaks (Fig. 3a). Mapping CSPs onto the structure of paenilamicin B2 revealed a well-defined region comprising the N-terminal half, with the strongest effect being located at position 6 of Aga (Fig. 3b and Supplementary Table 2). *N*-acetylpaenilamicin B2 also showed an additional cross-peak compared to paenilamicin B2, which we tentatively assigned to the methyl moiety of the newly attached acetyl group (Fig. 3a). Our data unequivocally demonstrated that PamZ mono-*N*-acetylates the N-terminal amino group at Aga-6 position of paenilamicin and thereby abolishes its antibacterial activity. Ultimately, this result is further supported by two synthetic diastereomers of paenilamicin B2 with L- instead of the native D-configuration at Aga-6 (**PamB2_1** and **PamB2_2**), that were both antibacterially less active[29] and that were not modified by PamZ (Fig. 4 and Supplementary Fig. 23).

**The structure of PamZ:acetyl-CoA binary complex.** A BLAST[30] search indicated that PamZ belongs to the GNAT superfamily with a sequence identity of 31% to the *N*-acetyltransferase, ZmaR, whose structure has not yet been determined and which confers resistance against the aminopolyol peptide antibiotic, zwittermicin A, in *Bacillus cereus* UW85 (Supplementary Fig. 24)[31]. We solved the crystal structure of PamZ in complex with acetyl-CoA at a resolution of 1.34 Å by using the uncharacterized *N*-acetyltransferase from *Streptococcus suis* 89/1591 (PDB 3G3S) for molecular replacement (Supplementary Table 3). The electron density was of excellent quality, allowed the modeling of the entire polypeptide chain, and unambiguously revealed the bound acetyl-CoA (Supplementary Fig. 25). PamZ comprises an N-terminal domain (NTD, residues 1–128, secondary structure elements indicated by primes) and a C-terminal domain (CTD,

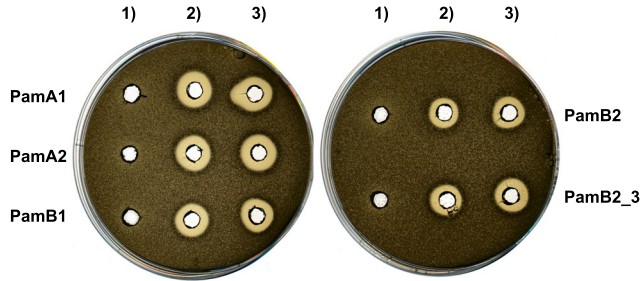

**Fig. 2 Deactivation of paenilamicins through PamZ-mediated *N*-acetylation tested by agar diffusion assay against *B. megaterium* as the indicator strain.** Paenilamicin variants (PamA1, A2, B1, B2) isolated from *P. larvae* and synthetic paenilamicin B2 (**PamB2_3**) were incubated in vitro with both acetyl-CoA and PamZ (1), acetyl-CoA only (2), or PamZ only (3). Samples 2 and 3 are negative controls and indicate the lack of bacterial growth.

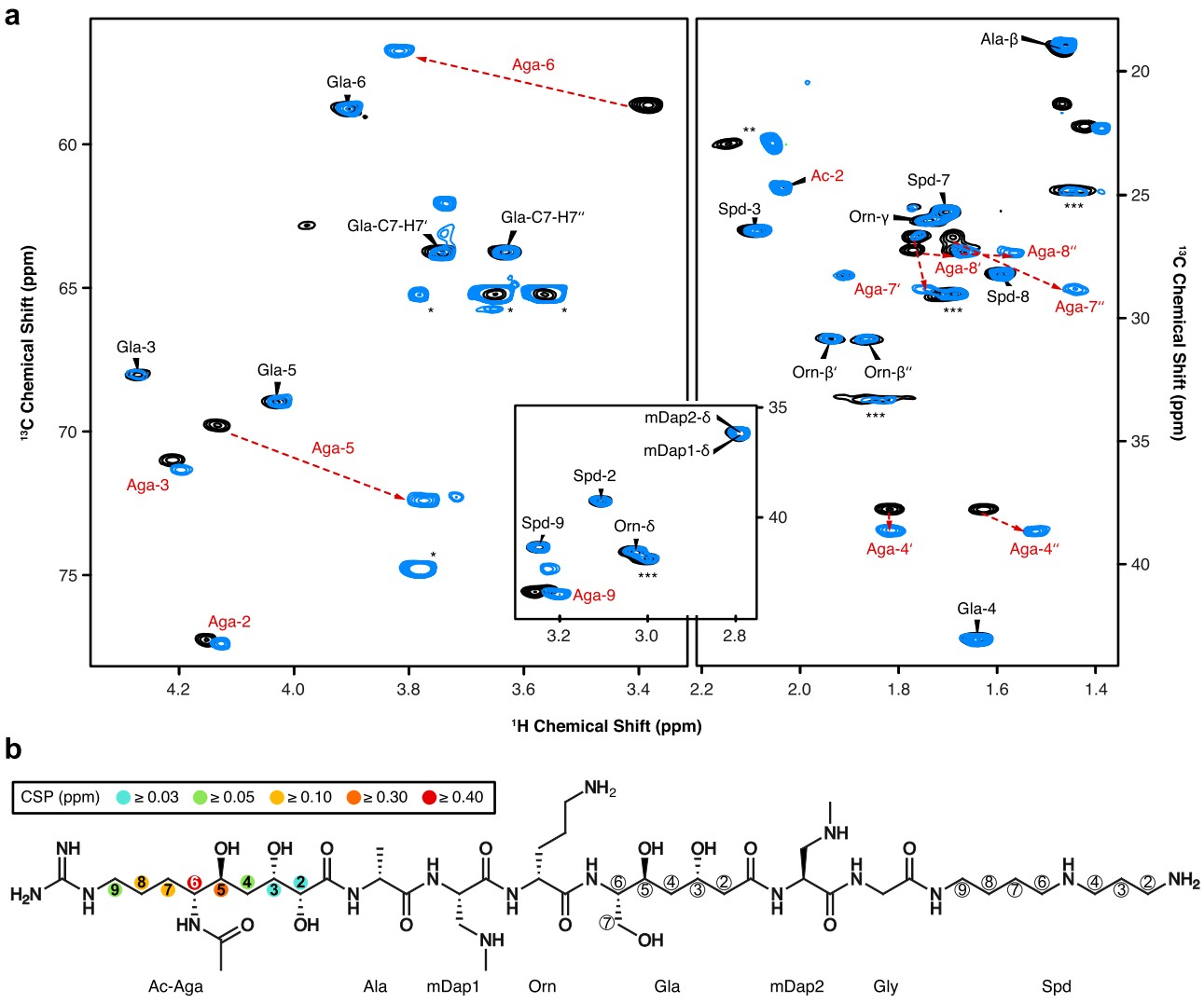

**Fig. 3 Identification of the *N*-acetylation site through 2D NMR spectroscopy. a** Overlay of relevant ¹H-¹³C HSQC sections of paenilamicin B2 (black) and *N*-acetylpaenilamicin B2 (blue). Strongly perturbed cross-peaks are highlighted with red labels. Known impurities are labeled with one, two and three asterisks arising from glycerol, acetic acid and residual purification traces of paenilamicin B1, respectively. The numbers (in backbone) and Greek letters (in amino acid residues) refer to atom positions in the corresponding building blocks as labeled in **b**. **b** Significant chemical shift perturbations (CSPs) and corresponding positions are indicated as circles in the chemical structure of *N*-acetylpaenilamicin B2 (see legend for color code). Aga - agmatinamic acid, Ala - alanine, mDap - *N*-methyldiaminopropionic acid, Orn - ornithine, Gla - galantinic acid, Gly - glycine, Spd - 4,3-spermidine, Ac - acetyl group.

residues 140–275), which both adopt the characteristic GNAT fold (Fig. 5a)[32]. The two tandem-GNAT domains, that may have originated from a gene duplication event, share low sequence identity (<20%) and are connected by an α-helical linker (α_bridge, residues 129–139). The overall fold of each domain is very similar to that of bacterial aminoglycoside *N*-acetyltransferases (AACs), as pairwise structural alignments with several AACs (PDB 1BO4, 1M4I, 1S3Z) gave root-mean-square deviations (RMSDs) of 2.9–4.2 Å for both the NTD and CTD (Supplementary Fig. 26)[33]. A structural superposition of the NTD and CTD of PamZ yielded an RMSD of 4.2 Å for 75 pairs of C_α atoms (Supplementary Fig. 27)[33].

However, a comparison with the typical GNAT fold revealed several unique features in PamZ. Instead of two N-terminal α-helices, α1 and α2, both domains of PamZ contain three short helical segments, α0-α1-α2 (α0'-α1'-α2'), which pack onto one face of the central antiparallel β-sheet, β2–β3–β4 (β2'–β3'–β4'), whereas helix α3 (α3') buries its other side. A kink in the backbone conformation of strand β3, involving residues T199 and C200, causes a strong right twist and thus a distortion of the

antiparallel β3–β4 arrangement, which led us to discriminate these strands as β3a/β3b and β4a/β4b (Fig. 5a). The central β-sheet is extended by strand β5' in the NTD, whereas the CTD shows the characteristic β-bulge of GNAT enzymes — a V-shaped cavity between strands β4b and β5 accommodates the pantetheine segment of CoA (Fig. 6a). Furthermore, the well-conserved pyrophosphate-binding loop (P-loop) of the GNAT family (R/Q-X-X-G-X-A/G)[26] is only present in the CTD of PamZ (Q-N-K-G-L-A) between strand β4b and helix α3 (Fig. 6a)[34], whereas the NTD is missing this signature motif. Accordingly, there is only one acetyl-CoA molecule canonically bound in the PamZ structure, namely to the CTD.

Hence, we concluded that the NTD is incompetent in binding acetyl-CoA and rather plays a structural role, in particular for substrate binding (see below). Notably, many GNAT enzymes exist as homodimers in solution with various arrangements of the monomer-monomer interface[32]. Likewise, AACs have often been crystallographically observed in a homodimeric state, although their quaternary structure in solution may vary[35]. PamZ exists as a monomer, both in solution and in the crystal (Supplementary

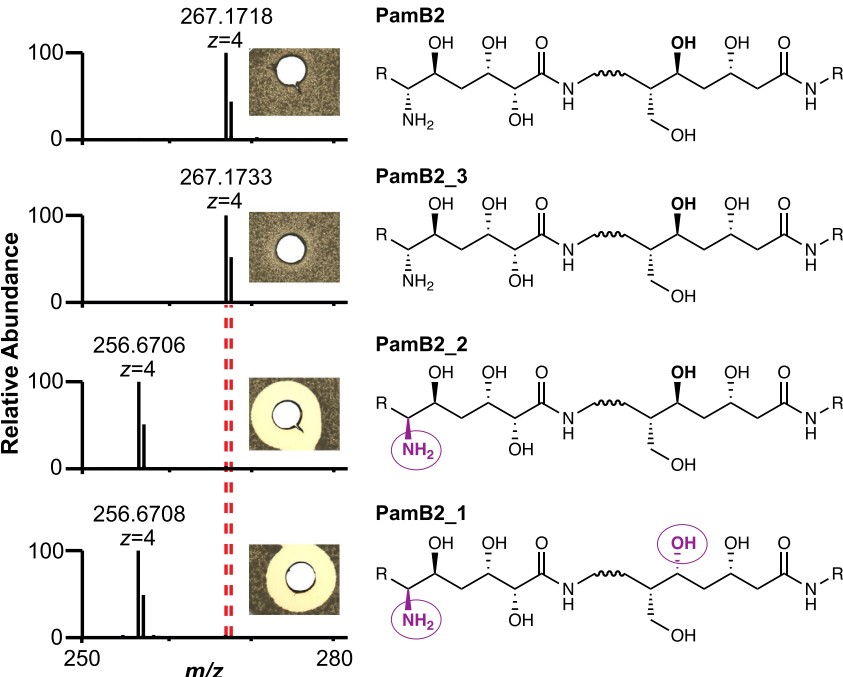

**Fig. 4 Substrate specificity and stereoselectivity of PamZ.** The natural product (**PamB2**), synthetic paenilamicin B2 (**PamB2_3**) and synthetic diastereomers of paenilamicin B2 (**PamB2_2**, **PamB2_1**) were incubated with PamZ and acetyl-CoA in vitro and tested in an agar diffusion assay against *Bacillus megaterium* (insets). From the chemical structure of paenilamicins, only the agmatinamic and galantinic acid are depicted to emphasize the changes in stereoconfiguration highlighted in purple and circles. Each single reaction was verified by HPLC-ESI-MS. Dashed lines indicate the mass shift of 42 Da (4 × 10.5 Da) due to *N*-acetylation.

Fig. 28). However, the tandem-GNAT domain constellation of PamZ achieves an intramolecular domain-domain interface that resembles that of some GNAT homodimers. There are several GNAT enzymes that utilize domain swapping of strand β6 to stabilize their homodimeric structure[34,36,37]. Interestingly, a major interface in PamZ is achieved by domain swapping of strand β6 (β6'), which inserts between strands β5' and β6' (β5 and β6) of the opposing domain and thus forms an extended, antiparallel, and strongly-twisted β-sheet throughout the enzyme (Fig. 5b). This β-sheet is only interrupted by the β-bulge in the CTD accommodating the cofactor and allowing the amide groups of its pantetheine portion to form pseudo-β-sheet hydrogen bonds to strand β4b (Fig. 6a). A very similar tandem arrangement of a pseudo-GNAT NTD and a canonical GNAT CTD can be found in the template protein (PDB 3G3S). Another example is the structure of mycothiol synthase MshD from *Mycobacterium tuberculosis*, which is also organized as a tandem repeat of two GNAT domains with a catalytically inactive NTD[38].

PamZ appears to utilize its NTD to form a well-defined substrate pocket with strands β5 and β6' representing its floor. A second interface between the NTD and CTD is accomplished through tight packing of helix α2' onto the small β3b–β4a sheet. Further interactions involve helix α2 of the CTD and the loops between α2' and β2' as well as β3' and β4' of the NTD. These inter-domain contacts fully cover the central groove that is normally found at the interface of homodimeric structures of GNAT enzymes and restrict substrate entry to the opening that is also used by the cofactor. This remaining cleft between the two domains of PamZ is decorated with several acidic residues (e.g. E89, E116, E118, D120, D162, D170, D215, E216, E217, E218, E272, E274, and the C-terminus) and thus deploys a large negatively charged surface to attract its polycationic substrate (Fig. 5c). A corridor that lies aside and beyond the acetyl group of the cofactor is approximately 7–8 Å deep and 8–9 Å wide with respect to the thioester carbonyl atom. Although we did not

obtain crystals of a ternary PamZ-acetyl-CoA-paenilamicin complex, the position of acetyl-CoA, the well-defined shape of the neighboring pocket and our knowledge about the substrate's N-terminal acetylation site allows us to predict that the Glm/Aga side chain of paenilamicin very likely penetrates into this pocket. Acidic residues D25 (loop between α1' and α2'), E122 (β6'), and E208 (β4a) are well-positioned within the pocket to accommodate and stabilize the guanidine group of Aga, as well as to tolerate the Nζ amine of Glm. Other residues that shape the substrate pocket include T58/T59 (loop between β3' and β4'), T98 (β5') and Y124 (β6') of the NTD as well as C200/Y201 (β3b) and S245/F247 (β5) of the CTD (Fig. 6c). This shows that both domains most likely contribute to substrate recognition. Moreover, the structure of PamZ explains its regioselectivity: if PamZ was to modify e.g. the terminal amino group of spermidine in paenilamicin, the enzyme would not require such a deep substrate-binding pocket. The architecture of the central groove between the NTD and CTD has evolved to optimally accommodate the N-terminal Glm/Aga building block of paenilamicin, whilst terminal amines such as those of spermidine, ornithine and lysine side chains would not occupy this binding pocket, as they would experience significantly less binding stabilization.

Such accommodation of Glm/Aga in the substrate pocket would position the N-terminal amino group of Aga-6 close to the thioester carbonyl of the cofactor. An active site aspartate or glutamate residue commonly acts as a general base to trigger the *N*-acetylation reaction by deprotonation of the amine followed by a nucleophilic attack at the carbonyl of the thioester[35]. In PamZ, the side chains of E122 (β6') as well as E208 (β4a) exhibit an interatomic distance of ~7 Å to the carbonyl atom of acetyl-CoA and thus might be in close proximity to the N-terminal amino group of Aga-6 (Fig. 6c). Residue S245 (β5) is sandwiched between E122 and E208, and may mediate deprotonation and/or proton shuttling. Furthermore, we cannot exclude the involvement of water molecules during proton transfer. An oxyanion

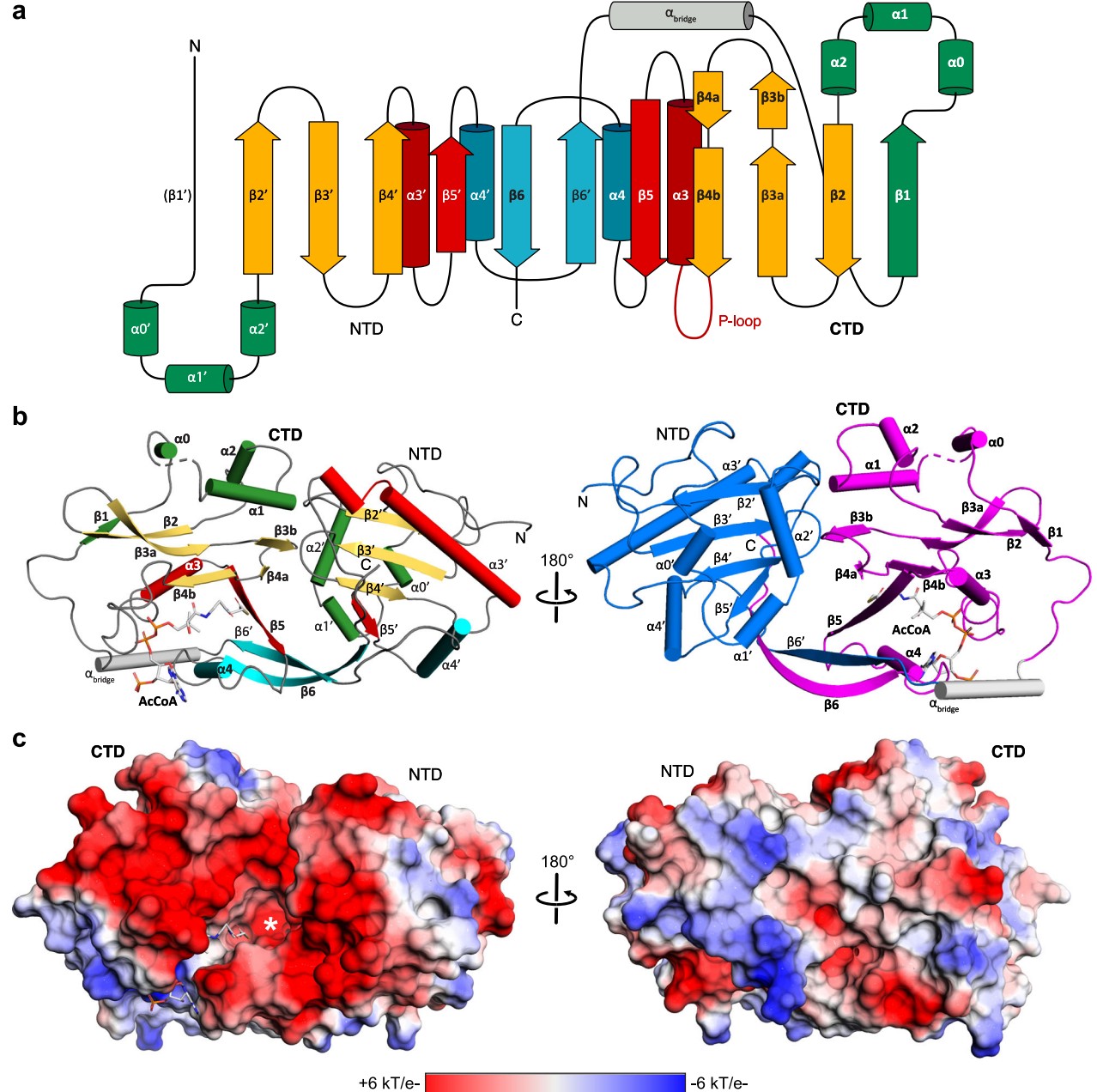

**Fig. 5 X-ray crystal structure of Gcn5-related *N*-acetyltransferase PamZ. a** Structural topology of PamZ with its characteristic tandem-GNAT fold. The protein structure is divided into an N-terminal (NTD) and a C-terminal (CTD) domain. Color coding of protein regions follows that of other bacterial GNATs, such as aminoglycoside *N*-acetyltransferases (AACs)[32]. **b** Cartoon representations of PamZ from two perspectives. The first perspective (left) follows the color code as in panel **a**. Acetyl-CoA (AcCoA) bound to the P-loop and β-bulge of the CTD is depicted as sticks. The β-bulge is formed by strands β4b and β5. The tandem-GNAT domains are highlighted (right) in blue (NTD) and purple (CTD). Secondary structure elements are labeled according to the protein topology. **c** Identical view as in panel **b** with the electrostatic potential mapped on the surface of PamZ, illustrating positive (blue) and negative (red) charges. The acetyl group attached to CoA (sticks) points into the active site highlighted by an asterisk. GNAT is an abbreviation of Gcn5-related *N*-acetyltransferase (Gcn5: general control non-repressed protein 5).

hole as described for myristoyl-CoA transferase[39] is not present in PamZ, but the amide proton of V211 (β4b) facilitates hydrogen bonding to the carbonyl oxygen of the thioester, which would increase the electrophilicity of the carbonyl carbon and stabilize the tetrahedral transition state after nucleophilic attack.

**Self-resistance mechanism of *P. larvae*.** The deactivation of paenilamicin through formation of *N*-acetylpaenilamicin by the action of PamZ (Supplementary Figs. 3–7) implicates that the

enzyme may confer self-resistance to the producer strain *P. larvae*. To test this hypothesis, we exposed the deletion mutant *P. larvae* Δ*pamZ* to a mixture of paenilamicin A1/B1 in an agar diffusion assay. The mixture, which was purified from *P. larvae* ERIC II, inhibited bacterial growth of the deletion mutant Δ*pamZ*, but not that of the WT strain (Fig. 7a).

This result demonstrated that *P. larvae* requires the resistance gene, *pamZ*, to protect itself from the deleterious effects of its own antibacterial agent, paenilamicin. For further experimental

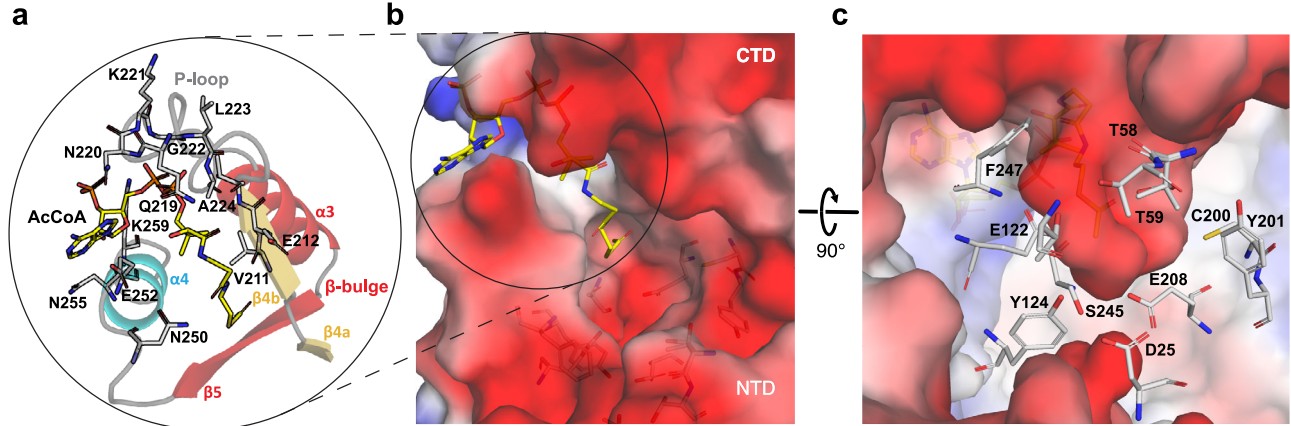

**Fig. 6 Active site of PamZ. a** Motifs A (β4–α3) and B (β5–α4) located in the C-terminal domain (CTD) interact with co-substrate acetyl-CoA. **b** Close-up view of the active site displaying the negatively charged groove (color code as in Fig. 5c). **c** Highlighted amino acid residues with hydrogen-donating and -accepting groups form the groove and are well-positioned to potentially interact with the substrate paenilamicin.

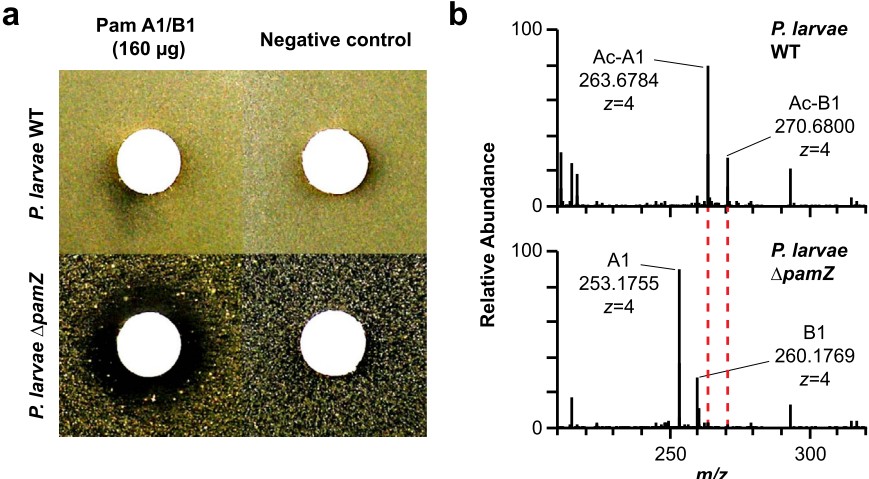

**Fig. 7 Self-resistance of *P. larvae* against paenilamicin. a** Deactivation of a paenilamicin mixture A1/B1 (left) was tested by an agar diffusion assay against *P. larvae* WT (top) and *P. larvae* Δ*pamZ* (bottom). The negative control (right) contained water only. **b** HPLC-ESI-MS spectra of cell lysates of *P. larvae* WT (top) and *P. larvae* Δ*pamZ* (bottom) are depicted. Relevant peaks for paenilamicin (A1/B1) and *N*-acetylpaenilamicin (Ac-A1/Ac-B1) species are labeled with corresponding *m/z* ratios ($z = 4$). WT - wild-type.

support, we analyzed supernatants and cell pellets of *P. larvae* WT and Δ*pamZ* for paenilamicins and *N*-acetylpaenilamicins. In cell lysates of *P. larvae* WT, we exclusively found *N*-acetylpaenilamicin, whereas for the deletion mutant Δ*pamZ* only unmodified paenilamicin (Fig. 7b) was detected. From paenilamicin isolates of the WT strain, primarily paenilamicin and only small amounts of *N*-acetylpaenilamicin were found in the supernatant by HPLC-ESI-MS (Supplementary Fig. 2). These findings demonstrate that the self-resistance factor PamZ enables *P. larvae* WT to acetylate and thus inactivate intracellular paenilamicin.

*N*-acetylation functions as an efficient self-protection mechanism by scavenging paenilamicin that reenters the cells of *P. larvae*. However, this mechanism may not apply to intracellular paenilamicin after its release from the NRPS-PKS assembly line. Instead, it seems very likely that an inactive precursor, i.e. a prodrug, of paenilamicin is produced to mask the strong antibacterial activity before cellular export. Along these lines, the biosynthetic gene cluster of paenilamicin[23] harbors the *pamJ* gene, which shows significant sequence similarity to a cyclic-peptide export ABC transporter with D-asparagine-specific peptidase activity that has been reported to be involved in a prodrug resistance mechanism in

nonribosomal peptide synthesis[40–43]. The peptidase recognizes and cleaves an *N*-acyl-D-asparagine unit of the prodrug. Accordingly, *P. larvae* must have developed a dual self-resistance mechanism against paenilamicin both potentially addressing the N-terminal Glm/Aga residue, specifically the N-terminal amino group at Aga-6 position, as modification site. Not only *P. larvae*, but also other bacteria belonging to the Firmicutes refer to a dual self-resistance mechanism associated with NRPS-PKS-derived compounds like amicoumacin[42,44], zwittermicin[31], and edeine[45] (Supplementary Fig. 29). In a very recent study, paenilamicin B2 showed an inhibitory effect ($IC_{50}$ of ~0.3 µM) on the *E. coli* ribosome in vitro, whereas the non-native diastereomer **PamB2_2** was approximately tenfold and the *N*-acetylpaenilamicin B2 approximately 100-fold less active[29]. The modifications of the N-terminal amino group at Aga-6 thus point to the importance of the N-terminal Glm/Aga residue as a major pharmacophore mediating recognition at the molecular target.

The insights into the pharmacophore region of paenilamicins and the structure of PamZ including its substrate-binding pocket may lead to the development of inhibitors against the self-resistance factor to weaken the bee larvae pathogen. This study lays an appreciable foundation to understand the self-resistance

mechanisms of *P.* larvae and to screen for potential small molecule inhibitors that target the active site of PamZ and compete with the naturally occurring paenilamicin. Consequently, the inhibitors would disable the self-resistance of *P. larvae* to paenilamicin resulting in the suicide of the bacterial pathogen for the benefit of the honey bee larvae. Such an anti-virulence strategy has the advantage that it circumvents the direct selection pressure on *P. larvae* to produce resistant strains because it only disarms the bacterium instead of interfering in essential bacterial functions. Therefore, it is not expected that inhibiting PamZ will readily lead to the development of resistant strains, as has already happened with the classical antibiotic treatment of AFB[46]. Assuming that the potential candidate has been confirmed in lab-scale applications, it could be applied in honey bee colonies by preventively impregnating the brood comb wax foundations with the substance so that it is dissolved in the brood food and then ingested by the larvae during early larval stages when *P. larvae* poses the most serious threat. Certainly, finding a small molecule inhibitor for PamZ and optimizing its application have to be further investigated in future studies. In summary, these results expand our knowledge of the molecular strategies exploited by *P. larvae* to survive in its ecological niche — knowledge that is needed to combat this pathogen and secure the health of bee colonies worldwide.

## Methods

**Bacterial strains and culture conditions**. The field strain *Paenibacillus larvae* (*P. larvae*) 04-309 (DSM 25430) and the deletion mutant 04-309 Δ*pamZ* were cultivated as follows: bacteria were grown on Columbia sheep blood agar (CSA, Thermo Fisher Scientific Oxoid, Schwerte, Germany) medium plates at 37 °C for 2–3 days. A preculture of 2 mL Mueller-Hinton-yeast-phosphate-glucose-pyruvate (MYPGP)[47] medium was inoculated with a single colony and grown overnight. A 50 mL culture of MYPGP broth was inoculated with the preculture to reach an optical density measured at 600 nm (OD$_{600}$) of 0.001. This main culture was incubated at 30 °C for 72 h with gentle shaking (80 rpm). Cultures were centrifuged at 3200×*g*, 4 °C for 30 min, and supernatants were stored at −20 °C until further use.

*Escherichia coli* (*E. coli*) BL21-Gold(DE3) cells were cultivated in Luria-Bertani (LB) medium at 37 °C and 180 rpm. The medium was supplemented with kanamycin (50 µg mL$^{−1}$) as an antibiotic based on the selection marker of the plasmid after transformation. Indicator strains like *B. megaterium* used for the agar diffusion assay were cultivated in LB medium at 37 °C and 180 rpm.

**Deletion mutant generation**. The generation of the *pamZ* deletion mutant was realized through a well-established protocol for *P. larvae* using the TargeTron Gene Knockout System (Sigma-Aldrich, Germany) based on group II intron insertion as previously described[11,14,15,24,48]. The *pamZ* gene of *P. larvae* DSM 25430 (GenBank CP003355.1, range from 1,729,003 to 1,729,830) was disrupted via site-specific insertion of a 900 bp-sized bacterial mobile group II intron LI.LtrB from *Lactococcus lactis* at position 118 from the start codon. The intron was previously modified to enable specific insertion at this site identified by a computer algorithm provided by the manufacturer (TargeTron™ Gene Knockout System, http://sigmaaldrich.com/crispr) with primers also identified by the computer algorithm (Supplementary Table 4)[49]. After successful cloning and transformation into *P. larvae* DSM 25430, screening for *P. larvae* DSM 25430 deletion mutants with the intron integrated in the *pamZ* gene was done via PCR with *pamZ*-specific primers (Supplementary Table 4 and Supplementary Fig. 30a). PCR analysis was performed with subsequent capillary gel electrophoresis (QIAxcel Advanced System with QIAxcel ScreenGel software v1.5.0.16, Qiagen, Hilden, Germany).

Growth of the *pamZ* deletion mutant in liquid MYPGP medium was not significantly altered in comparison to the wild-type strain (Supplementary Fig. 30b, two-way-ANOVA, *P* = 0.6486). In brief, growth curves were obtained as follows. *P. larvae* starting cultures had an optical density at 600 nm (OD$_{600}$) of 0.001 and were covered with mineral oil for anaerobic conditions. Cultures were grown in a 96-well-plate (Greiner Bio-One GmbH, Frickenhausen, Germany) and incubated at 37 °C while shaking in a Synergy HT plate reader with BioTek Gen5 software v1.10.8 (BioTek, Bad Friedrichshall, Germany). Measurements of the OD$_{600}$ took place hourly for 48 h. The experiment was repeated three times with three biological replicates with three technical replicates each. Representative results are shown. Statistical analysis was performed using GraphPad Prism software v6.07 (GraphPad Software, San Diego, CA, USA).

**Genomic DNA isolation**. Cells of *P. larvae* were picked from CSA plates, resuspended in 50 µL water and incubated at 95 °C for 10 min. They were centrifuged at

5000×*g* for 5 min and the supernatant containing the DNA was stored at −20 °C until further use. For gene amplification for cloning procedures, pure DNA was isolated by using the MasterPure™ Gram-Positive DNA Purification Kit (Epicentre, Illumina, San Diego, CA, USA) following the manufacturer's instructions.

**Plasmid construction and transformation**. Primers were designed for the amplification of the *pamZ* gene from *P. larvae* DSM 25430 with OligoAnalyzerTool (Integrated DNA Technologies) as well as Clone Manager 7 (Sci Ed Software) and purchased from Thermo Fisher Scientific (Supplementary Table 5). The gene *pamZ* was identified with the analysis tool antiSMASH 5.0[50], cloned into vector pET28a(+) introducing an N-terminal histidine-tag and a TEV site. Reactions were performed in the following conditions: initial denaturation at 95 °C for 5 min, followed by 30 cycles (105 s per cycle) at 98 °C for 30 s, at 61 °C for 30 s, and at 72 °C for 45 s, followed by a final extension step at 72 °C for 10 min. The PCR analysis was performed with the GelDoc System v0.2.14 (Intas Science Imaging Instruments GmbH). The amplicons were purified and digested with *Nhe*I and *Xho*I, ligated with the digested pET28a(+) vector, and transformed into *E. coli* BL21-Gold(DE3).

**Heterologous expression and protein purification**. Terrific broth (TB) medium was inoculated with an overnight culture of pET28a_pamZ transformed in *E. coli* BL21-Gold(DE3) cells to reach an OD$_{600}$ of 0.1 for the purification of PamZ. The culture was incubated at 37 °C and 180 rpm until OD$_{600}$ of 0.8–1.0. Expression was induced by addition of 0.2 mM (f.c.) isopropyl β-D-1-thiogalactopyranoside (IPTG). Cells were further incubated at 160 rpm, 18 °C for 20 h. Cells were harvested at 5000×*g*, 4 °C for 30 min (Beckman Coulter, Avanti J-26 XP with rotor JLA 8.1), and the pellet was resuspended in lysis buffer (500 mM sodium chloride, 50 mM TRIS/HCL pH 8.0, 20 mM imidazole). Then, 10 mM magnesium chloride, 5 µg mL$^{−1}$ DNase, 0.25 mg mL$^{−1}$ lysozyme, and 0.2 M benzamidine were added to the solution. The cell disruption was performed by the cell homogenizer at 15,000 psi (Constant Systems Ltd, United Kingdom). The cell lysate was centrifuged at 50,000×*g*, 4 °C for 30 min (Beckman Coulter, Avanti J-26 XP with rotor JA-25.50). The supernatant was loaded onto a His-Trap column using an ÄKTA protein purification system (ÄKTApurifier 10, GE Healthcare). The chromatography was run with a two-step gradient started with 100% starting buffer (500 mM sodium chloride, 50 mM TRIS/HCL pH 8.0, 20 mM imidazole) and switched to 50% elution buffer (500 mM sodium chloride, 50 mM TRIS/HCL pH 8.0, 250 mM imidazole) within 10 CV to elute the His$_6$-tagged PamZ. A His-Trap crude FF column (GE Healthcare) was used for this purification. Fractions of interest were collected and combined to increase protein concentration. Subsequently, TEV protease (1 mg per 10 mg of protein) was added to the concentrated protein solution and incubated at 4 °C for 16 h. The N-terminal, TEV-cleavable His$_6$-tag was separated from the untagged PamZ by a second nickel affinity chromatography. Size-exclusion chromatography was performed with a HiLoad 16/600 Superdex 75 pg column (GE Healthcare) to remove residual imidazole from the protein sample with buffer solution (150 mM sodium chloride, 20 mM TRIS/HCL pH 8.0). The flow rate was set to 1 mL min$^{−1}$. The chromatograms were recorded with Unicorn v5.20 (GE Healthcare). Fractions of interest were collected again, verified by SDS-PAGE and Coomassie staining, and then concentrated. Protein concentration was determined with the NanoPhotometer® P 330 (Implen, Munich, Germany). Protein samples were flash-frozen in liquid nitrogen and stored at −80 °C for further applications. The image of the gel was acquired with the software argusX1 v7.9.7 (Biostep) and the scanner system ViewPix 900 based on Epson scanner technology.

**Analytical size-exclusion chromatography**. Mixture A and B were used as standards. Mixture A contained aprotinin (3 mg mL$^{−1}$), carbonic anhydrase (3 mg mL$^{−1}$), conalbumin (3 mg mL$^{−1}$), and mixture B ribonuclease (3 mg mL$^{−1}$), ovalbumin (4 mg mL$^{−1}$). The chromatograms of mixture A and B were acquired as references to determine the oligomeric state of PamZ. Untagged PamZ (1.25 mg mL$^{−1}$) was prepared to obtain the best-fitted chromatogram. The size-exclusion chromatography was run with the ÄKTA protein purification system (ÄKTA-purifier 10, GE Healthcare), equipped with Superdex 75 10/300 GL and run with buffer solution (150 mM sodium chloride, 20 mM TRIS/HCL pH 8.0). The flow rate was set to 0.5 mL min$^{−1}$. The chromatograms were recorded with Unicorn v5.20 (GE Healthcare).

**Protein crystallization, structure determination, and refinement**. For crystallization experiments, PamZ was concentrated to 71 mg mL$^{−1}$. Crystallization was performed in a sitting drop vapor diffusion setup at 293 K. The reservoir solution was composed of 40% (w/v) PEG 3350, 50 mM ammonium sulfate, and 100 mM sodium acetate at pH 4.6. Prior to flash cooling, the crystals were cryo-protected in a reservoir solution supplemented with 20% (v/v) glycerol. Diffraction data were collected at beamline 14.2 at BESSY. Diffraction data were processed with XDS (Supplementary Table 3)[51]. The structure was solved by molecular replacement with PHASER 2.8.1[52] using the N-terminal domain of the PDB 3G3S. Since the C-terminal domain could not be readily placed the model was completed by Arp/wArp 8.0[53]. The structure was refined by maximum-likelihood restrained refinement in PHENIX 1.16_3549[54,55]. Model building and water picking were performed with COOT 0.8.1[56]. Hydrogen atoms for protein residues and ligands were

generated with PHENIX.REDUCE 3.7.201124[57]. Model quality was evaluated with MolProbity and the JCSG validation server (JCSG Quality Control Check v3.1)[58]. Figures were prepared using PyMOL (Schroedinger Inc.). Electrostatic potentials were calculated with APBS[59]. Structural alignments have been performed using SSM[33] integrated in COOT 0.8.1[56]. Structural homologs were identified with the DALI server[60]. Structural interfaces were analyzed with the PISA server (https://www.ebi.ac.uk/pdbe/prot_int/pistart.html)[61]. In addition, multiple sequence alignments were performed with ClustalOmega (https://www.ebi.ac.uk/Tools/msa/clustalo/) and visualized with Jalview v2.10.3b1 (https://www.jalview.org/).

**Compound isolation from the supernatant.** In all, 1 L of frozen supernatants of *P. larvae* ATCC 9545 or DSM 25430 cultures were thawed and then incubated with Amberlite XAD16 adsorption beads (1 g of beads per 10 mL culture filtrate, Sigma, St. Louis, MO, USA) and stirred for 16 h at room temperature. Then, the flow through was separated from the beads and a three-step gradient applied using 1 L of 10% (v/v) methanol followed by 1 L each of 90% (v/v) methanol and 90% (v/v) methanol plus 0.1% formic acid (f.c.) to finally obtain paenilamicin (and also *N*-acetylpaenilamicin). (*N*-acetyl)paenilamicin-containing fractions were concentrated and purified subsequently by using a Grace HPLC column (GROM-Sil 120 ODS-5-ST, 10 μm, 250 × 20 mm) coupled to an Agilent 1100 HPLC system (Agilent Technologies, Waldbronn, Germany) with a MWD UV detector. The separation was accomplished by a linear gradient elution using water plus 0.1% (v/v) formic acid as solvent A and acetonitrile plus 0.1% (v/v) formic acid as solvent B. The gradient started from 3% (v/v) to 15% (v/v) solvent B for 8 min, followed by 100% (v/v) solvent B for 7 min, and finished with an isocratic gradient of 100% (v/v) solvent B for 3 min. The flow rate was set to 20 mL min⁻¹. In the next step, (*N*-acetyl)paenilamicin-containing fractions were concentrated, adjusted with trifluoroacetic acid to approximately pH 2.0 to increase separation and purified by an Agilent HPLC column (PLRP-S, 100 Å, 10 μm, 150 × 25 mm) coupled to an Agilent 1100 HPLC system with a MWD UV detector for the separation of the native (*N*-acetyl)paenilamicin variants. (*N*-acetyl)paenilamicin was purified by using an isocratic gradient elution using water plus 0.1% (v/v) trifluoroacetic acid as solvent A and acetonitrile plus 0.1% (v/v) trifluoroacetic acid as solvent B. The isocratic gradient was started with 1% (v/v) solvent B for 8 min, followed by a linear gradient from 1% (v/v) to 95% (v/v) solvent B for 7 min, and finished with an isocratic gradient of 95% (v/v) solvent B for 5 min. The flow rate was set to 20 mL min⁻¹. (*N*-acetyl)paenilamicin-containing fractions were dried in vacuo, lyophilized to obtain pure compound, and verified by HPLC-ESI-MS/MS and ¹H-NMR spectroscopy.

**Compound extraction from cell pellet.** After cultivation of *P. larvae* DSM 25430 and its deletion mutant Δ*pamZ*, the cells were harvested and the cell pellets resuspended in 50% methanol (1 g per 2 mL solvent). The cells were disrupted by sonication (Branson Sonifier 250) for five cycles (15 s each cycle). In between each cycle, the cell lysate was incubated on ice for 60 s. The lysate was centrifuged at 5000 g, 15 °C for 30 min. The supernatant was analyzed for *N*-acetylpaenilamicin and paenilamicin by HPLC-ESI-MS.

**In vitro activation assay.** A reaction mixture consisted of 0.5 mM paenilamicin, 7.5 μM PamZ, 1 mM acetyl-CoA, 1.5 mM sodium phosphate buffer (pH 7.8). Also, samples were prepared each without enzyme and co-substrate as negative controls. The reaction mixture was incubated at 30 °C for 8 h. PamZ was removed by Amicon centrifugal filters (Merck KGaA, Germany) using a 10 kDa molecular weight cut-off filter. Deactivation of paenilamicin was tested against *B. megaterium* as indicator strain by agar diffusion assay and analyzed with HPLC-ESI-MS/MS and NMR spectroscopy. For the preparation of the ¹H-¹³C HSQC NMR experiment, excessive acetyl-CoA from the in vitro activation assay was removed by using an HPLC column (Phenomenex, Luna C18[2], 100 Å, 5 μm, 100 × 4.6 mm) coupled to an Agilent 1100 HPLC system (Agilent Technologies, Waldbronn, Germany) with a MWD UV detector. The separation was accomplished by a linear gradient elution using water plus 0.1% (v/v) formic acid as solvent A and acetonitrile plus 0.1% (v/v) formic acid as solvent B. The gradient started from 3% (v/v) to 15% (v/v) solvent B for 8 min, followed by 100% (v/v) solvent B for 2 min, and finished with an isocratic gradient of 100% (v/v) solvent B for 2 min. The flow rate was set to 0.6 mL min⁻¹.

For the determination of substrate specificity and stereoselectivity of PamZ including synthetic diastereomers, a reaction mixture consisted of 0.5 mM paenilamicin B2 (also for diastereomers), 7.5 μM PamZ, 1 mM acetyl-CoA, 1.5 mM sodium phosphate buffer (pH 7.8). Also, samples were prepared each without enzyme and co-substrate as negative controls. The reaction mixture was incubated at 30 °C for 2 h. PamZ was removed by Amicon centrifugal filters (Merck KGaA, Germany) using a 10 kDa molecular weight cut-off filter. After removal of the protein, the reaction mixture was tested against *B. megaterium* as indicator strain by agar diffusion assay and analyzed with HPLC-ESI-MS.

**Agar diffusion assay.** In all, 20 mL of LB medium including 0.75% (w/v) agar was inoculated with bacterial suspension of *B. megaterium* with a final OD₆₀₀ of 0.05. After solidification of the agar plate, holes were punched into the agar for activity testing. After sample preparation from in vitro activation assay, 10 μL of each sample was pipetted into the holes after the removal of PamZ, and the plate

incubated at 37 °C overnight. The agar plates were visualized with the scanner system ViewPix 900 based on Epson scanner technology.

**In vivo activation assay.** The growth of wild-type *P. larvae* DSM 25430 was compared to the growth of *P. larvae* DSM 25430 Δ*pamZ* in the presence of purified paenilamicin A1/B1 from bacteria supernatants in an agar diffusion assay. In brief, pre-cultures with 5 mL volume were grown in MYPGP broth at 37 °C while gently shaking overnight. Liquid lukewarm MYPGP agar was inoculated with *P. larvae* pre-cultures to result in a final optical density OD₆₀₀ of 0.05. Agar plates were poured and let harden. Meanwhile, 20 μL of paenilamicin A1/B1 dissolved in MilliQ (in total 160 μg per disk) were dispensed on filter disks and dried at room temperature. The dry filter disks were placed on the agar. The agar plates were incubated at 37 °C overnight. Clear zones of inhibition around the filter disks were indicative of a loss of paenilamicin resistance.

**Mass spectrometry analysis.** A 6530 Accurate-Mass Quadrupole Time-of-Flight (Q-TOF) LC/MS (Agilent Technologies, Waldbronn, Germany) was used to verify (*N*-acetyl)paenilamicin-containg fractions during the isolation and purification of paenilamicin. The Q-TOF was attached to an Agilent 1260 Infinity HPLC system and equipped with an HPLC column (Poroshell 120, EC-C8, 2.7 μm, 2.1 × 50 mm, Agilent Technologies, Waldbronn, Germany). The HPLC was started with a linear gradient from 5% (v/v) to 100% solvent B for 10 min using water plus 0.1% (v/v) formic acid as solvent A and acetonitrile plus 0.1% (v/v) formic acid as solvent B, followed by an isocratic gradient of 100% (v/v) for 1 min. The column was equilibrated with 5% (v/v) solvent B for 3 min. The flow rate was set to 0.5 mL min⁻¹. Other parameters were set as follows: positive mode, gas temperature to 300 °C, drying gas to 8 L min⁻¹, nebulizer to 35 psi, sheath gas temperature to 350 °C, sheath gas flow to 11 L min⁻¹, capillary voltage to 3500 V, fragmentor to 330 V, skimmer to 65 V, acquired rate to 1 spec s⁻¹. The MS data derived from the Q-TOF were acquired with MassHunter LC/MS Data Acquisition B.06.01 (Agilent Technologies), analyzed, and displayed with MassHunter Qualitative Analysis B.07.00 (Agilent Technologies).

An LTQ-Orbitrap XL hybrid ion trap-orbitrap (Thermo Fisher Scientific GmbH, Bremen, Germany) was used to verify the in vitro activation assays and to generate tandem mass spectra of paenilamicin and *N*-acetylpaenilamicin in data-dependent acquisition (DDA) mode. The LTQ-Orbitrap XL was attached to an analytical HPLC 1200 Infinity system (Agilent Technologies, Waldbronn, Germany) and equipped with an HPLC column (Poroshell 120, EC-C18, 2.7 μm, 2.1 × 50 mm, Agilent Technologies, Waldbronn, Germany). HPLC was run with a linear gradient using water plus 0.1% (v/v) formic acid as solvent A and acetonitrile plus 0.1% (v/v) formic acid as solvent B from 5% (v/v) to 100% (v/v) solvent B for 6 min, followed by an isocratic gradient of 100% (v/v) solvent B for 2 min. The column was equilibrated with 5% (v/v) solvent B for 2 min. The flow rate was set to 0.5 mL min⁻¹. The ESI source parameters were set as follows: product ion spectra were recorded in data-dependent acquisition (DDA) mode with a mass range from *m/z* 180 to *m/z* 2000 (MS1: FTMS, normal, 60,000, full, positive. MS2: FTMS, normal, 30000). The parameter for the DDA mode was set as follows: dynamic exclusion (repeat count: 3, repeat duration: 30 s, exclusion size list: 50, exclusion duration: 180 s), current scan event (minimum signal threshold: 10,000), activation (type: CID, default charge state: 2, isolation width: *m/z* 2.0, normalized collision energy: 35, activation Q: 0.25, activation time: 30 ms). The MS data derived from the LTQ-Orbitrap XL were acquired with XCalibur 2.2, analyzed, and displayed with QualBrowser (XCalibur 2.2, Thermo Fisher Scientific).

**Nuclear magnetic resonance spectroscopy.** NMR experiments were performed on a Bruker Avance III 700 MHz spectrometer equipped with a room-temperature TXI probe (Bruker, Karlsruhe, Germany). TopSpin 3.5 (Bruker, Karlsruhe, Germany) was used for data acquisition and processing. Spectra analysis was performed using NMRFAM-SPARKY[62,63]. ¹H and ¹H-¹³C HSQC spectra of paenilamicin and *N*-acetylpaenilamicin were recorded using samples in D₂O with 0.1% acetic acid-*d₄* at 298 K. ¹H-¹³C HSQC spectra were recorded with acquisition times of 120 ms and 9 ms in the direct ¹H and indirect ¹³C dimension, respectively. A delay Δ/2 of 1.72 ms was used for INEPT transfers corresponding to ¹J_{HC} of 145 Hz. Apodization of time-domain data was performed using a squared sine-bell function shifted by 90°. The 2D data was processed by applying linear forward prediction in the indirect ¹³C dimension and zero filling prior to Fourier transformation. ¹H chemical shifts were referenced externally using a sample of trimethylsilylpropanoic acid (TMSP-*d₄*, Deutero GmbH, Kastellaun, Germany) in D₂O with 0.1% acetic acid-*d₄* measured at 298 K. ¹³C chemical shifts were referenced indirectly using a correction factor of $f_{13C/1H} = 0.251449530$[64,65]. Chemical shift perturbations (CSPs) were calculated using the following equation[66]:

$$\text{CSP} = \sqrt{(f \times \Delta\delta_{13C})^2 + (\Delta\delta_{1H})^2} \qquad (1)$$

where $\Delta\delta_{13C}$ and $\Delta\delta_{1H}$ correspond to the ¹³C and ¹H chemical shift differences between paenilamicin B2 and *N*-acetylpaenilamicin B2 for each carbon-proton pair. We used a weighting factor $f$ of 0.06 to account for the much larger chemical shift dispersion in the ¹³C dimension (ca. 60 ppm) compared to that in the ¹H dimension (ca. 3.5 ppm).

**Chemical synthesis**. The total synthesis of paenilamicin B2 (**PamB2_3**) and its diastereomers (**PamB2_2**, **PamB2_1**) have been recently described[29].

**Reporting summary**. Further information on research design is available in the Nature Research Reporting Summary linked to this article.

## Data availability

The data that support this study are available from the corresponding author upon reasonable request. The crystallographic data generated in this study have been deposited in the Protein Data Bank under accession code 7B3A. Diffraction images have been deposited at www.proteindiffraction.org. The MS data generated in this study have been deposited in Mass spectrometry Interactive Virtual Environment (MassIVE) under project identifier MSV000088695 (https://doi.org/10.25345/C5J86H). Source data are provided with this paper.

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

## Acknowledgements

We thank the Deutsche Forschungsgemeinschaft (DFG, German Research Foundation) with SU 239/21-1 (project no. 279410221) for funding the project. This research was also funded by the Ministries responsible for Agriculture of the German Federal States of Brandenburg, Sachsen-Anhalt, Thüringen, Sachsen and the Senate of Berlin, Germany, as well as by the DFG, grant numbers GE1365/1-1, GE1365/1-2, and GE1365/2-1 to E.G. T.B. is grateful for the support of the DFG with RTG 2473 (Bioactive Peptides, project no. 392923329). We are grateful to Claudia Alings, Freie Universität Berlin, for help with crystallization. We acknowledge access to beamlines of the BESSY II storage ring (Berlin, Germany) via the Joint Berlin MX-Laboratory sponsored by Helmholtz-Zentrum Berlin für Materialien und Energie, Freie Universität Berlin, Humboldt-Universität zu Berlin, Max-Delbrück-Centrum, Leibniz-Institut für Molekulare Pharmakologie, and Charité-Universitätsmedizin Berlin.

## Author contributions

T.D., S.M., A.M., and R.D.S. designed the experiments. T.D., S.M., and R.S. purified paenilamicins and PamZ and also conducted the in vitro activation assays. T.D. set up the tandem-MS experiments and analyzed the MS data. B.L. performed the crystallization and elucidated the protein structure of PamZ. J.E. generated the deletion mutant *P. larvae* Δ*pamZ* and performed the in vivo activation assay of paenilamicin against wild-type and deletion mutant. T.B. and S.G. synthesized paenilamicin B2 and the two diastereomers. J.G. cultivated *P. larvae* wild-type and deletion mutant Δ*pamZ* and prepared the corresponding supernatants. A.M. acquired and analyzed the NMR data. T.D., M.C.W., E.G., A.M., and R.D.S. wrote the manuscript.

## Funding

## Competing interests

The authors declare no competing interests.
