## [Peer Review File · Nature Communications]

REVIEWER COMMENTS

Reviewer #1 (Remarks to the Author):

This is an interesting report, which focused on the self-resistance mechanism of the Bee pathogenic bacterium *Paenibacillus larvae*. The work could pave the way to disable the self-resistance of the bacterium to its own antibiotic, and would consequently render the bacteria defenseless, resulting in their suicide in favor of the bee hosts.

The authors' work is focused on sophisticated and convincing NMR studies to track down the exact acetylation site, which they showed caused the self-resistance.

The manuscript is lengthy, and most if not all experimental details could be placed into the Supporting files.

The authors also missed to comment on devastating bee loss due to phytochemicals, which is at least as devastating as the mentioned natural threats. It is important to list estimates to put the bacterial threat in context.

Minor errors: The authors missed to separate the numeral values from the degree centigrade symbol in most cases. Please correct.

Reviewer #2 (Remarks to the Author):

The authors have painstakingly carried out extensive characterisation of PamZ and its substrates and have notably identified the role and molecular details of the inactivation of the paenilamicin antibiotics that *P. larvae* produce to maintain their niche.

This manuscript is a solid and well-presented example of the utilisation of protein structure and biophysics to gain a detailed characterisation of an enzyme and its substrates. Although this pathogen only affects honeybees, the authors open with a very clear statement why this is an area of research that should be of interest to a broader readership.

As a whole, this body of work presents claims that are thoroughly substantiated through well executed experiments. I do not believe that any further experiments are required but the authors should take the following points into consideration.

A more detailed experimental exploration to support the assertions made around substrate binding based on your crystal structure could have been carried out. I.e. if you are pointing to residues that may be involved in substrate binding but you couldn't solve the structure with substrate bound, then mutagenesis and subsequent observation of the effects on activity would be suitable. The crystal structure has not really been used to its fullest in the manuscript but is nonetheless a good basis for future exploration.

Given you have access to a range of substrates and clearly plenty of enzyme, some interesting but straightforward enzymological experiments could have been carried out. Ellman's reagent (DTNB) makes following acetyltransferase reaction kinetics quite straightforward and could have been used to look for differences between the different paenilamicin species for example.

There are a couple sections that need clarification in the text to avoid confusion but appear to be unintentionally poorly worded:

1. In the identification of the location of acetylation position on the antibiotic, the text refers to a poorly defined nomenclature of Aga-x (where x is a number being referred to). This nomenclature is not readily apparent in the main text as figures 1 and 3b have chemical structures of the paenilamicin but no indication of which atoms this is referring to. The supplemental Table S2 has these numbers that also correspond to the colouring in Fig 3b. The reader should not have to delve 21 pages into the supplemental section to decipher this. Numbering the relevant atoms in one of the two main figures should be present to assist the reader.

2. The secondary structural elements (SSE) of the two domains have been differentiated by the use of primes (eg α_2' is the second alpha helix in the N-terminal domain). Lines 198-199 are referring to

SSEs in what I interpret as the N-terminal domains but do not have the primes there. This was confusing and should be rectified if that is indeed what is being referred to.

A few small typographical/stylistic errors should also be considered:

1. Line 39: the word “high” is not necessary here
2. Line 45: your team “solved” or “determined” the crystal structure, not “elucidated” it
3. Line 61: what does the acronym ERIC mean?
4. Line 186: you need a comma after the word “chain” otherwise this sentence is not completely clear
5. Line 375/376: the final concentrations of the additions of “magnesium chloride, DNase, lysozyme and benzamidine” would be helpful for a researcher wanting to reproduce your results.
6. Line 378: why state the exact model of centrifuge but not state what rotor was used. And similarly, more consistency in applying this rule would be beneficial as the subsequent line states an AKTA system but not which model.

Reviewer #3 (Remarks to the Author):

Reviewer comments for manuscript NCOMMS-21-44898, entitled “Molecular Basis of Antibiotic Self-Resistance in a Bee Larvae Pathogen”.

The authors studied bacterial self-resistance to the hybrid NRPS/PKS antimicrobial peptide, paenilamicin, produced by *Paenibacillus* larvae – a deadly Gram-positive pathogen which causes widespread disease in honey bees. It was discovered that co-production of the N-acetyltransferase PamZ (within same biosynthetic gene cluster as paenilamicin) led to the deactivation of paenilamicin in *P. larvae* as a protective mechanism. PamZ supplementation could also provide protection against paenilamicin in other bacteria (e.g. *Bacillus megaterium*), but only if the substrate acetyl-CoA was also supplemented. Notably, the N-acetylation site was convincingly identified in this study and the crystal structure of PamZ (in complex with acetyl-CoA) was also elucidated at 1.34 Å resolution.

Overall, this is very impressive work and the authors comprehensively demonstrated their findings. I believe the manuscript will be of great interest to a broad scientific audience, especially those studying infectious disease of honey bees. I have listed several, mostly minor, comments and suggestions below. The only component that I feel is lacking in the manuscript is a balanced discussion on the biological relevance of the findings within a broader context. Specifically, how the discoveries from this work could be functionally applied to improve honey bee disease management at the field level. As described below, expanding on some of the discussion points could be extremely helpful for enhancing the interpretability and overall applicability of the findings across disciplines.

My comments and suggestions are as follows:

- Line 77-79: It is stated that “...only *P. larvae* can be isolated as a pure culture from AFB-diseased larval cadavers, while other microbial competitors are absent in the degradation process of the infected larvae” with a 1945 article from Holst cited to support this statement. However, this statement is factually incorrect based on more modern evidence which show advanced culturing techniques are able to isolate many organisms besides *P. larvae* from infected honey bee larvae. Exemplifying this, standard methods for American foulbrood research explicitly indicate the necessity of heat treatment of clinical field samples to eliminate microbial competitors which acts to select for heat-resistant *P. larvae* spores on isolation agar (de Graaf et al. 2013, *J Apicult Res*). While paenilamicin can undoubtedly act as a defence molecule against microbial competitors (and is the primary point being made earlier in the sentence), the latter part of this sentence should be corrected.

- Line 116: I can appreciate the usefulness of a standard indicator strain (such as the *Bacillus megaterium* used in this study), however, I am curious as to why more relevant honey bee microbes naturally present in the larval gut were not tested. For example, *Lactobacillus apis* and *Apilactobacillus kunkeei* are symbionts commonly found in healthy larval guts and are similarly closely related (phylogenetically speaking) to *P. larvae*. Is paenilamicin A/B not effective against these health-associated symbionts? Perhaps they can produce their own N-acetyltransferases which may provide

protection against *P. larvae*. From an evolutionary perspective, this might be a worthwhile point to comment on given the role of the commensal microbiota in preventing *P. larvae* infection.

- Line 282: It is unclear why only paenilamicin A1 & B1 (from *P. larvae* ERIC II) were tested in the self-resistance assays, but not paenilamicin A2 & B2 (from *P. larvae* ERIC I). A clarification statement would be helpful since both were evaluated in the earlier results sections.

- Line 317-321: This section should be preceded by a “Concluding remarks” header. Moreover, the article seemingly ends abruptly without much discussion on the biological relevance of the findings within a broader context. Stating the findings will help “...combat this pathogen and secure health of bee colonies worldwide” does not meaningfully contribute to idea generation on how the study findings could be translated for functional improvement of honey bee health at the field level. It would be helpful to lead the reader in this section and provide some knowledge gaps that are still unanswered and future studies that are needed in order to advance our ability to target PamZ. How do you envision these discoveries will ultimately impact disease management practices in apiculture? Is it feasible to supplement a small molecule inhibitor of PamZ to honey bee larvae, and how would this functionally be delivered to the hive? Addressing these topics would also help to add more generalizable interest to the article and disseminate the research findings across multiple scientific disciplines.

POINT-BY-POINT RESPONSE

Reviewer #1 (Remarks to the Author):

Comment 1:

This is an interesting report, which focused on the self-resistance mechanism of the Bee pathogenic bacterium *Paenibacillus larvae*. The work could pave the way to disable the self-resistance of the bacterium to its own antibiotic, and would consequently render the bacteria defenseless, resulting in their suicide in favor of the bee hosts. The authors' work is focused on sophisticated and convincing NMR studies to track down the exact acetylation site, which they showed caused the self-resistance.

Response 1:

We thank reviewer 1 for his/her comments and for highlighting the interest of the manuscript.

Comment 2:

The manuscript is lengthy, and most if not all experimental details could be placed into the Supporting files.

Response 2:

We agree with you and placed the experimental section into the supplementary information.

Comment 3:

The authors also missed to comment on devastating bee loss due to phytochemicals, which is at least as devastating as the mentioned natural threats.

Response 3:

Thank you for raising this essential point in terms of the impact on bee loss by chemicals. There are no studies showing that phytochemicals induce devastating bee losses. However, might the reviewer refer to pesticides instead of phytochemicals? While there are numerous studies implicating devastating effects of pesticides on individual bees in laboratory assays, there is no convincing study showing pesticides can kill entire colonies. Quite the contrary is true: all studies trying to kill colonies by applying field relevant doses of pesticides failed to demonstrate colony losses (Cresswell *et al. Ecotoxicology* **2011**, *20*, 149). The reason is that honey bee colonies come into contact with pesticides during foraging. Hence, a healthy colony can buffer temporarily occurring, massive losses of individual forager bees by increasing brood rearing in the brood season. The queen simply lays more eggs and the nurse bees simply raise more larvae.

Nevertheless, we agree with you that pesticides are potential threats for bees. Therefore, we have embedded a sentence in the introduction: "In order to secure human food supply, it is therefore important to ensure the health of honey bees, which is continuously threatened by the overuse of insecticides such as neonicotinoid (Henry *et al. Science* **2012**, *336*, 348-350) in agriculture and also by various viral, bacterial and fungal pathogens as well as metazoan parasites."

Comment 4:

It is important to list estimates to put the bacterial threat in context.

Response 4:

Thank you for this essential comment and for raising the significance of the bacterial threat against honey bees. We have added the following sentence to the introduction: "AFB is the most serious bacterial disease of honey bees and is classified as notifiable disease in most countries because it is highly contagious and lethal to entire colonies. Furthermore, most authorities consider killing of diseased colonies and burning of the hive material the only workable control measure resulting in considerable economic losses in apiculture."

Comment 5:

Minor errors: the authors missed to separate the numeral values from the degree centigrade symbol in most cases. Please correct.

Response 5:

Thank you for spotting these mistakes. We have corrected these by adding spaces in between. Since the methods section have been completely moved to the supplementary information that is not going to be edited in terms of the formatting instructions of *Nature Communications*, we have not highlighted the changes.

Reviewer #2 (Remarks to the Author):**Comment 6:**

The authors have painstakingly carried out extensive characterisation of PamZ and its substrates and have notably identified the role and molecular details of the inactivation of the paenilamicin antibiotics that *P. larvae* produce to maintain their niche. This manuscript is a solid and well-presented example of the utilisation of protein structure and biophysics to gain a detailed characterisation of an enzyme and its substrates. Although this pathogen only affects honeybees, the authors open with a very clear statement why this is an area of research that should be of interest to a broader readership. As a whole, this body of work presents claims that are thoroughly substantiated through well executed experiments. I do not believe that any further experiments are required but the authors should take the following points into consideration.

Response 6:

We thank reviewer 2 for the appreciation of the relevance of this work.

Comment 7:

A more detailed experimental exploration to support the assertions made around substrate binding based on your crystal structure could have been carried out. I.e. if you are pointing to residues that may be involved in substrate binding but you couldn't solve the structure with substrate bound, then mutagenesis and subsequent observation of the effects on activity would be suitable. The crystal structure has not really been used to it fullest in the manuscript but is nonetheless a good basis for future exploration. Given you have access to a range of substrates and clearly plenty of enzyme, some interesting but straightforward enzymological experiments could have been carried out. Ellman's reagent (DTNB) makes following acetyltransferase reaction kinetics quite straightforward and could have been used to look for differences between the different paenilamicin species for example.

Response 7:

Thank you for your advice on future experiments. We absolutely agree with you that performing the above-mentioned experiments would make the characterization of the enzyme complete and would support the crystal structure even more. We will take your advice into consideration for our future work.

Comment 8:

There are a couple sections that need clarification in the text to avoid confusion but appear to be unintentionally poorly worded:

1. In the identification of the location of acetylation position on the antibiotic, the text refers to a poorly defined nomenclature of Aga-x (where x is a number being referred to). This nomenclature is not readily apparent in the main text as figures 1 and 3b have chemical structures of the paenilamicin but no indication of which atoms this is referring to. The supplemental Table S2 has these numbers that also correspond to the colouring in Fig 3b. The reader should not have to delve 21 pages into the supplemental section to decipher this. Numbering the relevant atoms in one of the two main figures should be present to assist the reader.

Response 8:

Thank you for your attention and observation. It is truly beneficial for a better understanding by defining the position of Aga-x in one of both figures. We kindly appreciate this comment and have defined the numbering in Figure 3b.

Comment 9:

2. The secondary structural elements (SSE) of the two domains have been differentiated by the use of primes (e.g. $\alpha 2'$ is the second alpha helix in the N-terminal domain). Lines 198-199 are referring to SSEs in what I interpret as the N-terminal domains but do not have the primes there. This was confusing and should be rectified if that is indeed what is being referred to.

Response 9:

Thank you for spotting this sentence. We used the term “N-terminal α -helices” to describe the starting point of both domains (NTD and CTD). To avoid confusion, we have added the secondary structure elements of NTD in brackets:

“Instead of two N-terminal α -helices, $\alpha 1$ and $\alpha 2$, both domains of PamZ contain three short helical segments, $\alpha 0$ - $\alpha 1$ - $\alpha 2$ ($\alpha 0'$ - $\alpha 1'$ - $\alpha 2'$), which pack onto one face of the central antiparallel β -sheet, $\beta 2$ - $\beta 3$ - $\beta 4$ ($\beta 2'$ - $\beta 3'$ - $\beta 4'$), whereas helix $\alpha 3$ ($\alpha 3'$) buries its other side.”

Comment 10:

A few small typographical/stylistic errors should also be considered:

1. Line 39: the word “high” is not necessary here.

Response 10:

Thank you for spotting this word. We have removed the word “high” in this sentence.

Comment 11:

2. Line 45: your team “solved” or “determined” the crystal structure, not “elucidated” it.

Response 11:

Thank you for spotting this word. We have replaced the word “elucidated” by “determined”.

Comment 12:

3. Line 61: what does the acronym ERIC mean?

Response 12:

Thank you for highlighting this word. ERIC is the acronym for Enterobacterial Repetitive Intergenic Consensus. In this case, we have added an introductory sentence in the main text to introduce the acronym.

“The use of enterobacterial repetitive intergenic consensus (ERIC) sequence primers has revealed four well-described genotypes ERIC I to ERIC IV for *P. larvae* which differ in virulence on the larval and colony level as well as in pathogenesis strategies employed to kill the host.”

Comment 13:

4. Line 186: you need a comma after the word “chain” otherwise this sentence is not completely clear.

Response 13:

Thank you for the grammar correction. We have put a comma after the word “chain”.

Comment 14:

5. Line 375/376: the final concentrations of the additions of “magnesium chloride, DNase, lysozyme and benzamidine” would be helpful for a researcher wanting to reproduce your results.

Response 14:

Thank you for highlighting this sentence. We absolutely agree with you and added the final concentrations of used substances for reproducibility. Since we have moved the entire method section to the supplementary information that is not going to be edited in terms of the formatting instructions of *Nature Communications*, we have not highlighted the changes in the supplementary information but write it below:

“Then, 10 mM magnesium chloride, 5 $\mu\text{g mL}^{-1}$ DNase, 0.25 mg mL^{-1} lysozyme and 0.2 M benzamidine were added into the solution.”

Comment 15:

6. Line 378: why state the exact model of centrifuge but not state what rotor was used. And similarly, more consistency in applying this rule would be beneficial as the subsequent line states an AKTA system but not which model.

Response 15:

Thank you for this comment. We have included more information in terms of the rotor and ÄKTA system in the text. Since we have moved the entire method section to the supplementary information that is not going to be edited in terms of the formatting instructions of *Nature Communications*, we have not highlighted the changes in the supplementary information but write it below:

“Cells were harvested at 5000 g, 4 °C for 30 min (Beckman Coulter, Avanti J-26 XP with rotor JLA 8.1) and...”

“Cell lysate was centrifuged at 50000 g, 4 °C for 30 min (Beckman Coulter, Avanti J-26 XP with rotor JA-25.50).”

“Supernatant was loaded onto a His-Trap column using an ÄKTA protein purification system (ÄKTApurifier 10, GE Healthcare Life Sciences).”

Reviewer #3 (Remarks to the Author):**Comment 16:**

Reviewer comments for manuscript NCOMMS-21-44898, entitled “Molecular Basis of Antibiotic Self-Resistance in a Bee Larvae Pathogen”. The authors studied bacterial self-resistance to the hybrid NRPS/PKS antimicrobial peptide, paenilamicin, produced by *Paenibacillus larvae* – a deadly Gram-positive pathogen which causes widespread disease in honey bees. It was discovered that co-production of the N-acetyltransferase PamZ (within same biosynthetic gene cluster as paenilamicin) led to the deactivation of paenilamicin in *P. larvae* as a protective mechanism. PamZ supplementation could also provide protection against paenilamicin in other bacteria (e.g. *Bacillus megaterium*), but only if the substrate acetyl-CoA was also supplemented. Notably, the N-acetylation site was convincingly identified in this study and the crystal structure of PamZ (in complex with acetyl-CoA) was also elucidated at 1.34 Å resolution.

Overall, this is very impressive work and the authors comprehensively demonstrated their findings. I believe the manuscript will be of great interest to a broad scientific audience, especially those studying infectious disease of honey bees.

Response 16:

Thank you for the acknowledgement and appreciation of our work.

Comment 17:

I have listed several, mostly minor, comments and suggestions below. The only component that I feel is lacking in the manuscript is a balanced discussion on the biological relevance of the findings within a

broader context. Specifically, how the discoveries from this work could be functionally applied to improve honey bee disease management at the field level.

Response 17:

We like the idea of a balanced discussion on the biological relevance and highly appreciate you bringing this up. This work establishes a basis to develop small molecule inhibitors of PamZ as candidate for establishing an anti-virulence strategy against *P. larvae* infection. By blocking the self-resistance mechanisms of *P. larvae* against paenilamicin, this secondary metabolite would kill *P. larvae*. We agree that such an anti-virulence strategy could contribute to a sustainable treatment of AFB and, thus, to improve honey bee disease management in the field. We amended the conclusion section accordingly (please see our answer to comment 21).

Comment 18:

As described below, expanding on some of the discussion points could be extremely helpful for enhancing the interpretability and overall applicability of the findings across disciplines.

My comments and suggestions are as follows:

Line 77-79: It is stated that "...only *P. larvae* can be isolated as a pure culture from AFB-diseased larval cadavers, while other microbial competitors are absent in the degradation process of the infected larvae" with a 1945 article from Holst cited to support this statement. However, this statement is factually incorrect based on more modern evidence which show advanced culturing techniques are able to isolate many organisms besides *P. larvae* from infected honey bee larvae. Exemplifying this, standard methods for American foulbrood research explicitly indicate the necessity of heat treatment of clinical field samples to eliminate microbial competitors which acts to select for heat-resistant *P. larvae* spores on isolation agar (de Graaf et al. 2013, *J. Apicult. Res.*). While paenilamicin can undoubtedly act as a defence molecule against microbial competitors (and is the primary point being made earlier in the sentence), the latter part of this sentence should be corrected.

Response 18:

Thank you for highlighting this sentence. However, to our opinion this is clearly a misunderstanding of the existing literature. For instance, the publication (de Graaf et al., *J. Apicult. Res.* **2013**) does not refer to "clinical field samples" that need to be treated by heat but to "biological samples" containing *P. larvae* spores. In this context, biological samples are brood comb honey, adult bees etc. but not the ropy mass (containing vegetative bacteria) and not the foulbrood scales (containing only spores of *P. larvae*). Heat treatment to biological samples is rather not applied to kill other spores in the samples but to enhance spore germination of *P. larvae* (Forsgren et al., *Vet. Microbiol.* **2008**).

However, we agree that the wording of the sentence was indeed not optimal. We were referring to the larval cadaver degraded to a ropy mass at the end of the pathogenic process where no saprophytic competitors are present. Therefore, we have re-written the sentence as follows: "It is currently assumed that paenilamicin is produced as a defense molecule against microbial competitors, since only *P. larvae* can be isolated as a pure culture from the cadavers of AFB-killed larvae, while other saprophytic competitors are absent in the degradation process of the larval cadavers to the characteristic ropy mass (Holst, *Science* **1945**)."

Comment 19:

Line 116: I can appreciate the usefulness of a standard indicator strain (such as the *Bacillus megaterium* used in this study), however, I am curious as to why more relevant honey bee microbes naturally present in the larval gut were not tested. For example, *Lactobacillus apis* and *Apilactobacillus kunkeei* are symbionts commonly found in healthy larval guts and are similarly closely related (phylogenetically speaking) to *P. larvae*. Is paenilamicin A/B not effective against these health-associated symbionts? Perhaps they can produce their own N-acetyltransferases which may provide protection against *P. larvae*. From an evolutionary perspective, this might be a worthwhile point to comment on given the role of the commensal microbiota in preventing *P. larvae* infection.

Response 19:

Thank you for your comment and interest in the microbial interaction/competition in the larval midgut. We are fully aware of the complexity of possible interactions between *P. larvae* and other microbes

present in the larval gut or in the brood cell. Therefore, we already screened the activity of paenilamicin against a panel of bacteria and fungi associated with honey bee colonies or regularly isolated from brood comb honey (Garcia-Gonzalez *et al.*, *MicrobiologyOpen* **2014**). From this panel of tested bacteria, we chose *B. megaterium* as indicator strain for our study at hand. We also already tested the activity of paenilamicin against *B. thuringiensis*, (Bulatov *et al.* *JACS* **2022**), also often found in honey bee colonies, and against the bee-associated saprophyte *P. alvei* commonly found in the larval gut and responsible for the degradation of larvae killed by *Melissococcus plutonius* (causative agent of European Foulbrood) (Müller *et al.* *Angew. Chem. Int. Ed.* **2014**).

The two bacterial species mentioned by the reviewer rather belong to the microbiota of adult bees: *Apilactobacillus kunkeei* colonizes the honey crop of adult bees and *Lactobacillus apis* was isolated from the stomach of adult bees. The latter was shown to have an *in vitro* inhibitory effect on the growth of *P. larvae* (AFB) and *M. plutonius* (EFB). Testing a mixture of Lactobacilli in experimentally infected larvae revealed a slight though statistically significant reduction in larval mortality (Olofsson and Vasquez **2008**; Forsgren *et al.* **2010**) suggesting the general possibility that pathogenesis of AFB could be influenced by feeding probiotic bacteria. However, even in the Lactobacilli treated group of infected larvae 50 % died from infection, indicating that *P. larvae* infection could not be prevented.

While the microbiota of the adult honey bee has attracted much attention in the recent past leading to major improvements in knowledge of both its composition and its function (Vásquez *et al.* **2012**; Raymann and Moran **2018**), data on the larval microbiota is still contradictory and the role of the microbiota during development and for disease resistance is poorly understood.

Culture-based work suggested that the larval stages are devoid of a constant microbiota (Gilliam and Prest **1987**; Engel and Moran **2013**; Raymann and Moran **2018**). A view that was supported by a study presenting data obtained from surface-sterilized larvae via non-culture-based methods which showed a very limited or possibly absent gut microbiota in honey bee larvae (Martinson *et al.* **2012**). Consequently, these authors suggested that honey bee larvae remain mainly sterile until adult bee emergence or at least lack a defined microbiota (Martinson *et al.* **2012**). Other studies reported similar results (Mohr and Tebbe **2006**; Vojvodic *et al.* **2013**; Hroncova *et al.* **2015**) and detected only few bacterial groups (Acetobacteraceae, *Lactobacillus* sp.) in the larval gut microbiota of managed *A. mellifera*. Especially early larval stages (L1-L3), which are attacked by *P. larvae*, mostly lacked mutualistic bacteria. Even the recently identified honey bee host-specific microorganisms (*Snodgrassella alvi*, *Gilliamella apicola*, *Frischella perrara*) were not consistently found in L1 to L3 larvae (Hroncova *et al.* **2015**).

Considering this lack of robust knowledge on the larval microbiota, we postponed screening the activity of paenilamicin against microbes presumably occurring in the larval midgut until we have finished our own studies on the microbiota in the larval gut and the brood cell. We fully agree that addressing the interactions between *P. larvae* and other bacteria – if present – in the larval midgut is a very interesting topic which will therefore be part of our future studies on host-microbiota-pathogen interactions during AFB pathogenesis. However, such studies would clearly be beyond the scope of the study at hand.

Comment 20:

Line 282: It is unclear why only paenilamicin A1 & B1 (from *P. larvae* ERIC II) were tested in the self-resistance assays, but not paenilamicin A2 & B2 (from *P. larvae* ERIC I). A clarification statement would be helpful since both were evaluated in the earlier results sections.

Response 20:

Thank you for this comment. This work should only focus on the *P. larvae* ERIC II, which produces all four paenilamicin variants but preferably paenilamicin A1/B1 and A2/B2 in fairly small amounts (Fig. S2). Therefore, paenilamicin A1/B1 was used for the self-resistance assay in terms of the natural abundance and occurrence produced by *P. larvae* ERIC II. We only mentioned and used *P. larvae* ERIC I in the context of isolation and purification purposes since paenilamicin variants A2 and B2 were produced by *P. larvae* ERIC I in higher amounts. The full set of paenilamicin variants were then only used for substrate specificity and biological activity of PamZ *in vitro*.

Comment 21:

Line 317-321: This section should be preceded by a “Concluding remarks” header. Moreover, the article seemingly ends abruptly without much discussion on the biological relevance of the findings within a broader context. Stating the findings will help “...combat this pathogen and secure health of bee colonies worldwide” does not meaningfully contribute to idea generation on how the study findings could be translated for functional improvement of honey bee health at the field level. It would be helpful to lead the reader in this section and provide some knowledge gaps that are still unanswered and future studies that are needed in order to advance our ability to target PamZ. How do you envision these discoveries will ultimately impact disease management practices in apiculture? Is it feasible to supplement a small molecule inhibitor of PamZ to honey bee larvae, and how would this functionally be delivered to the hive? Addressing these topics would also help to add more generalizable interest to the article and disseminate the research findings across multiple scientific disciplines.

Response 21:

We highly appreciate your interest in this work since it is of high interest to find a solution approach for honey bee colony losses due to AFB. We really like the idea to discuss potential applications to honey bee health at field level and highlight the importance of threats to bees and the ecosystem. Therefore, we have added the following lines to the end of the manuscript as concluding remarks:

“This study lays a good foundation to understand self-resistance mechanisms of *P. larvae* and to screen for potential small molecule inhibitors that target the active site of PamZ and compete with the naturally occurring paenilamicin. Consequently, the inhibitors would disable the self-resistance of *P. larvae* to paenilamicin resulting in suicide of the bacterial pathogen for the benefit of the honey bee larvae. Such an anti-virulence strategy has the advantage that it circumvents the direct selection pressure on *P. larvae* to produce resistant strains because it only disarms the bacterium instead of interfering in essential bacterial functions. Therefore, it is not expected that inhibiting PamZ will readily lead to the development of resistant strains, as has already happened with classical antibiotic treatment of AFB (Miyagi *et al.* 2000). Assuming that the potential candidate has been confirmed in lab-scale applications, it could be applied in honey bee colonies by preventively impregnating the brood comb wax foundations with the substance so that it is dissolved in the brood food and then ingested by the larvae during early larval stages when *P. larvae* poses the most serious threat. Certainly, finding a small molecule inhibitor for PamZ and optimizing its application have to be further investigated in future studies.”

REVIEWER COMMENTS

Reviewer #1 (Remarks to the Author):

Reviewer #2 (Remarks to the Author):

I believe the responses to the reviewers comments have been satisfactorily addressed and modifications to the text reflect this.

It reads much better as a result. I don't have anything further to add.

Reviewer #3 (Remarks to the Author):

The authors have adequately addressed all concerns that I specified in the first round of revisions. I believe this work is now acceptable for publication.

POINT-BY-POINT RESPONSE

We thank all reviewers for the appreciation of our work and also for the time and effort to review our work as well as for the inspiring comments and discussions.